# Effect of Urease Inhibitors and Nitrification Inhibitors Combined with Seaweed Extracts on Urea Nitrogen Regulation and Application

**Furong Xiao [1,2], Dongpo Li [1,]*, Lili Zhang [1], Yandi Du [3], Yan Xue [1], Lei Cui [1,2], Ping Gong [1], Yuchao Song [1], Ke Zhang [1,2], Yiji Zhang [1,2], Yonghua Li [4], Jinming Zhang [5] and Yongkun Cui [5]**

1    Institute of Applied Ecology, Chinese Academy of Sciences, Shenyang 110016, China
2    University of Chinese Academy of Sciences, Beijing 100049, China
3    Chaoyang County Agricultural Technology Extension Center, Chaoyang 122000, China
4    North Huajin Chemical Industries Group Corporation, Panjin 124021, China
5    Jinxi Natural Gas Chemical Co., Ltd., Huludao 125001, China
*    Correspondence: lidp@iae.ac.cn

**Abstract:** In order to improve the application effect of inhibitors, the combination of seaweed extracts (SE), urease inhibitors (UIs) and nitrification inhibitors (NIs) were added to urea fertilizers for providing a theoretical basis to develop the efficient stabilized nitrogen (N) fertilizer. The combinations were tested in outdoor pots with no N fertilizer (CK) and application of urea alone (U) as control, and SE, N-butyl phosphorothioate triamine (NBPT), 3,4-dimethylpyrazole phosphate (DMPP), 2-chloro-6-trimethylpyridine (CP), and combinations of SE with each of the three inhibitors were added to urea to make seven fertilizer prototypes. The results showed that the addition of inhibitors and SE could regulate the N transformation, enzyme activity and microbial biomass carbon (MBC) contents and significantly improve maize yield and N use efficiency (NUE) ($p < 0.05$). Compared with the addition of inhibitor alone, the addition of SE was beneficial to N uptake by plants at the early growth stage of maize, but reduced the inhibitors effect; DMPP + S significantly increased the maize total biomass and NUE ($p < 0.05$), and the grain yield, N uptake and NUE by 15.6%, 28.5% and 30.6%, respectively. Therefore, the addition of DMPP with SE to urea significantly improved yield when planting maize in loess areas.

**Keywords:** biostimulants; urea; nitrogen transformation; nitrogen use efficiency

## 1. Introduction

Application of nitrogen (N) is an important step to ensure high crop yields [1]. According to the United Nations, global crop production increased by about 250% from 1965 to 1998, however, the amount of N fertilizer applied also increased by about 430% [2]. In China, from the 1970s to the 2020s, total crop production and yield per unit area increased by 98% and 71%, respectively, but the application of N fertilizer also increased by 271% [3]. The excessive input of N into agroecosystems not only increases production costs, but also causes many environmental problems [4]. More than half of the N applied to agroecosystems is not used by plants [5], and most of the N is lost to the environment through ammonia volatilization, nitrate leaching, and denitrification [6], causing various environmental problems, such as soil acidification [7] and the increase of the greenhouse effect [8]. Therefore, improving N use efficiency (NUE) and reducing N fertilizer application is a global issue and a key to modernize agricultural production in China, and the development of new efficient fertilizers is considered to be an effective measure to improve NUE.

N transformation inhibitors include urease inhibitors (UIs) and nitrification inhibitors (NIs), among which UIs generally inhibit urease activity in soils, thus delaying the hydrolysis of Urea-N [9], and NIs generally inhibit nitrification by inhibiting the activity of

nitrosobacteria or ammonia monooxygenase (AMO) [10,11]. In general, we define fertilizers with inhibitors as stabilized fertilizers. Previous studies have shown that the addition of UIs and NIs to urea fertilizers is effective in reducing N losses [12–14], and the addition of inhibitors is considered to have positive environmental benefits [15]. However, N that is not lost to the environment by application of inhibitors is not fully absorbed by the plant, and the effect of the addition of inhibitor alone on increasing crop yield is limited [16,17]. Meanwhile, as scientists continue to study inhibitors, it is found that the effect of inhibitors is affected by various factors in the soil, such as pH [18], organic matter [19], water-holding capacity (WHC) [20], texture [21], and microbial community structure [22], and that there are limits to the application of inhibitors under different environmental conditions. Therefore, the effect of applied inhibitor alone is not fully effective to ensure increase and stabilization of grain yield, and develop a new generation of efficient stabilized fertilizer products, which is an inevitable trend in the future development of stabilized fertilizers.

Recently, with the development of biostimulants, the application of biostimulants has been increasing in the agricultural field has become increasing and better known. Unlike traditional biofertilizers, biostimulants do not promote the growth of plants by providing them with nutrients, but by regulating their physiological metabolic functions and nutrient uptake capacity through the hormones and other components contained in them [23,24]. As the largest class of bio-resources in the ocean, seaweeds contain a large number of nutrients that are lacking in terrestrial ecosystems [25]. Seaweed extracts (SE) are rich in growth hormones, alginate, polysaccharides laminaran and other polysaccharides, and SE plays a key regulatory role in plant growth [26]. In current agricultural production, SE is mostly made into plant growth products or mixed with traditional pesticides and foliar fertilizers [27], which have been shown to promote seed germination [28], development of plant root elongation and yield formation [25], improve stress tolerance in plants [29] and fruit quality [30]. Therefore, we would like to solve the problem of limited yield increase of inhibitors by combining biostimulants with inhibitors to develop a new generation of stabilized fertilizers. However, it is unknown whether the combination of biostimulants with inhibitors can further promote plant growth and increase yield compared to traditional stabilized fertilizers, and whether the combination of them can enhance the effect of the inhibitors. This study aims to improve the application effect of traditional stabilized fertilizers and provide a theoretical basis for the development of a new generation of efficient stabilized fertilizers.

## 2. Materials and Methods

### 2.1. Experimental Site and Soils

A pot experiment was carried out at the National Field Observation and Research Station of Agroecosystems in Shenyang, Liaoning province (41°31′ N, 123°24′ E), in which Dongdan-6531 spring maize (*Zea mays* L., a late maturing maize variety based on Food and Agriculture Organization of the United Nations (FAO) standards, with a 121 to 130 days growing period, grown from May to October 2020) was planted. The mean annual air temperature is 7–8 °C, and the mean annual precipitation is approximately 700 mm. The frost-free period is 147–164 days. Loess samples (0–0.20 m) were collected from Xianyang City (107°48′ E, 34°59′ N) in Shanxi Province of northwest China with the physicochemical properties shown in Table 1.

**Table 1.** Physicochemical properties of loess (0–0.20 m soil layers).

| Soil Type | Texture Class | pH | Organic Matter (g/kg) | Total N (g/kg) | $NH_4^+$-N (mg/kg) | $NO_3^-$-N (mg/kg) | Total Phosphorus (g/kg) | Available Phosphorus (mg/kg) | Total Potassium (g/kg) | Available Potassium (mg/kg) |
|---|---|---|---|---|---|---|---|---|---|---|
| loess | loam | 8.01 | 31.26 | 1.14 | 9.17 | 10.70 | 0.67 | 11.45 | 40.45 | 257.37 |

## 2.2. Experimental Design

The soil selected for this experiment is black loessial soil, which originated from loess and is mainly distributed in northwest China, commonly known as loess. The soil samples collected from 0–0.20 m layer were mixed thoroughly, and after removed the fine root debris, they were laid flat on a plastic film for use. Nine treatments were established with three replications each: (1) no N fertilizer (CK); (2) urea (U); (3) urea + SE (S); (4) urea + NBPT (NBPT); (5) urea + DMPP (DMPP); (6) urea + CP (CP); (7) urea + NBPT + SE (NBPT + S); (8) urea + DMPP + SE (DMPP + S); (9) urea + CP + SE (CP + S). Each pot contains 6 kg of air-dried soil. The fertilizers urea, triple superphosphate and potassium chloride were applied at a rate of 0.7 g N, 0.12 g $P_2O_5$ and 0.15 g $K_2O$ per kg soil, respectively. The application rates of SE, NBPT, DMPP and CP were 6‰, 0.25%, 0.5%, and 0.25% [29,31,32], respectively on the w/w basis of urea. The SE and inhibitors applied to each treatment were weighed and mixed thoroughly with urea, then the prepared urea fertilizer was mixed with the soil, the fertilizer was mixed with the soil again by folding diagonally for several times, and the mixed soil was transferred into pots (diameter 26 cm and height 28 cm, cross-section is trapezoidal, plastic material), followed by packing and irrigated to maintain a constant moisture content of 60% field capacity. Watering was carried out every day during maize growth, and no topdressing during the growth period of maize. Five seeds were sown in each pot, and thinned to one plant per pot after germination and seedling establishment. Urea was supplied by China National Pharmaceutical Group Corporation, containing 46% N. Triple superphosphate was supplied by Yunnan Tianhua Group Co, containing 43% $P_2O_5$. Potassium chloride was produced by Russia, containing 60% $K_2O$. SE was supplied by a Chinese company, pH 7.4, containing 40% alginate, 7% polysaccharides laminaran, 30.2% organic matter, 4.5% moisture. NBPT was supplied by Macklin Biotechnology, with a purity of 97%. DMPP and CP were supplied by Maya Reagent Biotechnology, with a purity of 97% and 98%, respectively.

## 2.3. Soil and Maize Sampling

Soil samples were collected at maize seedling stage (growth stage 15 in BBCH-scale, 38 days after planting), elongation stage (growth stage 25 in BBCH-scale, 65 days after planting), filling stage (growth stage 75 in BBCH-scale, 102 days after planting), and maturity stage (growth stage 99 in BBCH-scale, 135 days after planting) using a 5-point sampling method, with three replicates for each treatment. The collected soil samples were mixed well, removed fine roots, debris, etc., and sieved through 2 mm sieve and set aside. The leaf area and chlorophyll content of the maize plant were measured at the silking stage (growth stage 67 in BBCH-scale) by YMJ-B and CCM-200 instruments, respectively, and the average value was calculated. At maturation, the whole maize plant in pot were collected, and the maize kernels, stalks and roots were collected and air-dried to determine the grain yield, total biomass and root biomass of maize. Samples of maize kernels, stalks and roots were also collected for analysis.

## 2.4. Soil and Plant Analyses

Soil pH was measured using a pH meter in a 1:2.5 soil deionized water suspension [18]. Soil organic matter content was determined by oxidation with potassium dichromate and titration with ferrous ammonium sulfate [33]. Total N of soil was determined by dry combustion with an elemental analyzer (Carlo Erba CNS analyzer NA 1500 series 2, Germany) [34]. Soil total phosphorus (P) was digested by $HClO_4$ and available P was extracted with 0.5 mol $L^{-1}$ $NaHCO_3$, then both analyzed by the molybdenum blue method [35,36]. Soil total potassium (K) was digested by HCl and available K was extracted with 1 mol $L^{-1}$ $NH_4OAc$ and determined by the flame photometer method [36,37].

Soil Urea-N was extracted with KCL-PMA (2 mol $L^{-1}$ KCL and 5 mg $L^{-1}$ PMA) [38], $NH_4^+$-N and $NO_3^-$-N were extracted with 2 mol $L^{-1}$ KCL [39], and measured on a continuous flow analyzer (AAIII, Norderstedt, Germany).

Determination of urease activity with reference to the method of Tabatabai [40], a certain amount of urea solution was added to the soil sample and incubated for 5 h at 37 °C. The residual Urea-N was measured and the amount of urea hydrolyzed per unit time was calculated to characterize the soil urease activity, and was expressed as mg $kg^{-1}$ $h^{-1}$. Determination of potential nitrification rate (PNR) with reference to the method of Hart [41], a certain amount of ammonium sulfate solution was added to the soil sample and incubated for 5 h at 25 °C, the transformation of nitrite nitrogen ($NO_2^-$-N) to $NO_3^-$-N was inhibited by sodium chlorate, and the $NO_2^-$-N released during incubation was extracted with KCL and measured at 520 nm. The $NO_2^-$-N production per unit time was calculated to represent the PNR, and was expressed as mg $kg^{-1}$ 5 $h^{-1}$. Determination of nitrate reductase (NR) activity with reference to the method of Abdelmagid [42], a certain amount of potassium nitrate was added to the soil sample and incubated for 24 h at 25 °C under water, the nitrite reductase activity was inhibited by 2,4-dinitrophenol reagent, and the $NO_2^-$-N released during incubation was extracted with KCL and measured at 520 nm. The $NO_2^-$-N production per unit time was calculated to represent the NR activity, and was expressed as mg $kg^{-1}$ 24 $h^{-1}$. For determination of MBC with reference to the method of Liu [43], the 15 g soil sample was fumigated for 24 h at 25 °C in a vacuum desiccator containing 50 mL of $CHCl_3$, the fumigated and a nonfumigated control sample were both extracted with 100 mL of 0.5 mol $L^{-1}$ $K_2SO_4$ on a reciprocal shaker for 60 min, MBC was extracted with $K_2SO_4$, and measured on a total C analyzer (TOC Analyzer, Multi C/N 3000, Analytik Jena, Germany), the difference in extractable C contents between the fumigated and non-fumigated samples was divided by 0.45 to calculate MBC.

All plant samples were dried at 65 °C until constant weight to calculate total biomass of maize, then ground and sieve through a 250-μm mesh for the analysis of total N using an elemental analyzer (VARIO MACRO cube).

*2.5. Calculations and Statistical Analyses*

Nitrification inhibition rate (%) was calculated using the following [32]:

$$\text{Nitrification inhibition rate (\%)} = (a - b)/a \times 100\%$$

where: a, the $NO_3^-$-N of soil applied urea only (mg/kg), b, is the $NO_3^-$-N of soil applied with biostimulant, UIs, NIs (mg/kg).

$$\text{Shoot/root ratio} = (\text{total biomass} - \text{root biomass})/\text{root biomass}$$

NUE were calculated by using formulas as follows [44]

$$\text{NUE} = (Y - YC)/NF$$

where: Y, plant N uptake with N fertilizer; YC, plant N uptake with no fertilizer; NF, the amount of N fertilizer applied.

Multiple comparisons among the treatments were explained using Tukey test. Differences were considered significant at $p < 0.05$. All statistical analyses were performed using Microsoft Excel 2010 (Redmond, WA, United States) and IBM SPSS 21.0 (Chicago, IL, United States). Graphs were prepared with Origin 2021. The data in the tables are the average value $\pm$ standard error.

## 3. Results

*3.1. The Transformation of N*

3.1.1. The Contents of Urea-N in Soils

At the seedling stage, no Urea-N was founded in the soils of all treatments, indicating that urea was fully hydrolyzed. Therefore, soil Urea-N was not tested for the three subsequent stages.



3.1.2. The Contents of $NH_4^+$-N in Soils

At the seedling stage, the $NH_4^+$-N contents of the applied CP were significantly higher than the other treatments, CP had 255.5 mg kg$^{-1}$ of $NH_4^+$-N, which was significantly higher than CP + S (Table 2, $p < 0.05$). At the elongation stage, the $NH_4^+$-N contents of CP was significantly higher than the other treatments ($p < 0.05$), which showed that the addition of CP was beneficial to the retention of $NH_4^+$-N in loess and CP had positive application effect. At the filling stage, the $NH_4^+$-N of all the applied N treatments was lower than CK, and after the filling stage, the $NH_4^+$-N contents of each treatment tended to stabilize (Table 2).

**Table 2.** Contents of $NH_4^+$-N of different treatments in soil (mg kg$^{-1}$).

| Treatments | Seedling (15) | Elongation (25) | Filling (75) | Maturity (99) |
|---|---|---|---|---|
| CK | 18.8 ± 2.8 c | 13.7 ± 0.24 b | 16.1 ± 2.2 a | 15.9 ± 1.7 a |
| U | 15.1 ± 1.4 c | 13.8 ± 0.6 b | 10.2 ± 0.4 bc | 16.4 ± 2.3 a |
| S | 16.9 ± 2.7 c | 12.3 ± 0.7 b | 7.8 ± 0.6 c | 9.5 ± 0.8 bc |
| NBPT | 17.0 ± 1.9 c | 12.9 ± 2.3 b | 10.5 ± 1.1 bc | 8.2 ± 1.0 c |
| DMPP | 18.0 ± 1.4 c | 14.4 ± 0.4 b | 13.5 ± 2.0 ab | 11.8 ± 1.1 b |
| CP | 255.5 ± 22.7 a | 22.2 ± 2.0 a | 9.7 ± 1.8 bc | 10.8 ± 0.4 bc |
| NBPT + S | 16.8 ± 3.2 c | 14.7 ± 1.7 b | 11.1 ± 1.1 bc | 11.0 ± 0.5 bc |
| DMPP + S | 15.3 ± 1.6 c | 14.2 ± 2.9 b | 13.1 ± 0.7 ab | 17.0 ± 0.4 a |
| CP + S | 93.8 ± 12.2 b | 15.3 ± 0.4 b | 9.8 ± 0.1 bc | 10.8 ± 0.7 bc |

Values represented mean ± standard deviations ($n = 3$). Different lowercase letters in the column mean a significant difference between treatments (Tukey test, $p < 0.05$). Treatments: CK: no N fertilizer; U: urea; S: urea + seaweed extracts; NBPT: urea + N-butyl phosphorothioate triamine; DMPP: urea + 3,4-dimethylpyrazole phosphate; CP: urea + 2-chloro-6-trimethylpyridine; NBPT + S: urea + N-butyl phosphorothioate triamine + seaweed extracts; DMPP + S: urea + 3,4-dimethylpyrazole phosphate + seaweed extracts; CP + S: urea + 2-chloro-6-trimethylpyridine + seaweed extracts. The same as below.

3.1.3. The Contents of $NO_3^-$-N in Soils

The soil $NO_3^-$-N contents of each treatment tended to decrease with the growth process of maize, while the highest $NO_3^-$-N contents were at the seedling stage (Table 3). At the seedling stage, the $NO_3^-$-N contents of applied N treatments were significantly higher than CK, where the $NO_3^-$-N contents of S was 532.1 mg kg$^{-1}$, which was significantly higher than the other treatments, and the addition of SE all caused a decreased in $NO_3^-$-N contents compared to the addition of NBPT, DMPP, and CP alone (Table 3), with a decrease of 40.7%, 10.3% and 69.4%, respectively. The addition of SE significantly increased the $NO_3^-$-N contents compared to U ($p < 0.05$). Among the applied inhibitor alone treatments, $NO_3^-$-N contents of NBPT were always higher than the other treatments from the seedling stage to elongation stage, with 4.5 mg kg$^{-1}$ of $NO_3^-$-N contents at the elongation stage, which was significantly higher than the other treatments ($p < 0.05$). The $NO_3^-$-N contents of each treatment increased at the filling stage, and then tended to stabilize at the maturity stage (Table 3).

**Table 3.** Contents of $NO_3^-$-N of different treatments in soil (mg kg$^{-1}$).

| Treatments | Seedling (15) | Elongation (25) | Filling (75) | Maturity (99) |
|---|---|---|---|---|
| CK | 15.5 ± 0.6 g | 1.9 ± 0.1 def | 2.3 ± 0.1 d | 3.6 ± 0.2 a |
| U | 298.8 ± 28.9 c | 2.0 ± 0.0 def | 2.8 ± 0.2 d | 3.8 ± 0.2 a |
| S | 532.1 ± 22.8 a | 1.5 ± 0.4 f | 6.4 ± 0.5 b | 2.7 ± 0.1 b |
| NBPT | 363.2 ± 10.2 b | 4.5 ± 0.1 a | 6.3 ± 0.1 b | 2.6 ± 0.3 b |
| DMPP | 258.0 ± 2.6 d | 2.8 ± 0.2 bc | 6.3 ± 0.4 b | 3.7 ± 0.2 a |
| CP | 267.2 ± 3.9 cd | 2.5 ± 0.2 cd | 2.4 ± 0.1 d | 3.4 ± 0.3 a |
| NBPT + S | 215.4 ± 7.9 e | 3.2 ± 0.4 b | 8.3 ± 0.1 a | 2.6 ± 0.1 b |
| DMPP + S | 231.5 ± 7.5 de | 2.4 ± 0.2 cde | 6.2 ± 0.4 b | 3.4 ± 0.3 a |
| CP + S | 81.7 ± 5.3 f | 1.7 ± 0.1 ef | 4.4 ± 0.2 c | 2.3 ± 0.2 b |

Different lowercase letters in the column mean a significant difference between treatments (Tukey test, $p < 0.05$). Treatments: CK: no N fertilizer; U: urea; S: urea + seaweed extracts; NBPT: urea + N-butyl phos-phorothioate triamine; DMPP: urea + 3,4-dimethylpyrazole phosphate; CP: urea + 2-chloro-6-trimethylpyridine; NBPT + S: urea + N-butyl phosphorothioate triamine + seaweed extracts; DMPP + S: urea + 3,4-dimethylpyrazole phos-phate + seaweed extracts; CP + S: urea + 2-chloro-6-trimethylpyridine + seaweed extracts.

### 3.2. Soil Nitrification Inhibition Rate of Different Treatments at the Seedling Stage

The effect of the inhibitors diminished with the growth process of maize, thus the soil nitrification inhibition rate at the seedling stage was calculated in this study (Figure 1). Both NBPT and SE had positive effect on soil nitrification. Compared to the applied inhibitor alone, the application with inhibitors and SE significantly increased the soil nitrification inhibition rate, and the combination of both decreased the soil inorganic N ($NH_4^+$-N and $NO_3^-$-N) contents (Tables 2 and 3), while the $NH_4^+$-N contents of CP + S were significantly lower than CP (Table 2, $p < 0.05$), thus CP + S would reduce the inhibitory effect of CP.

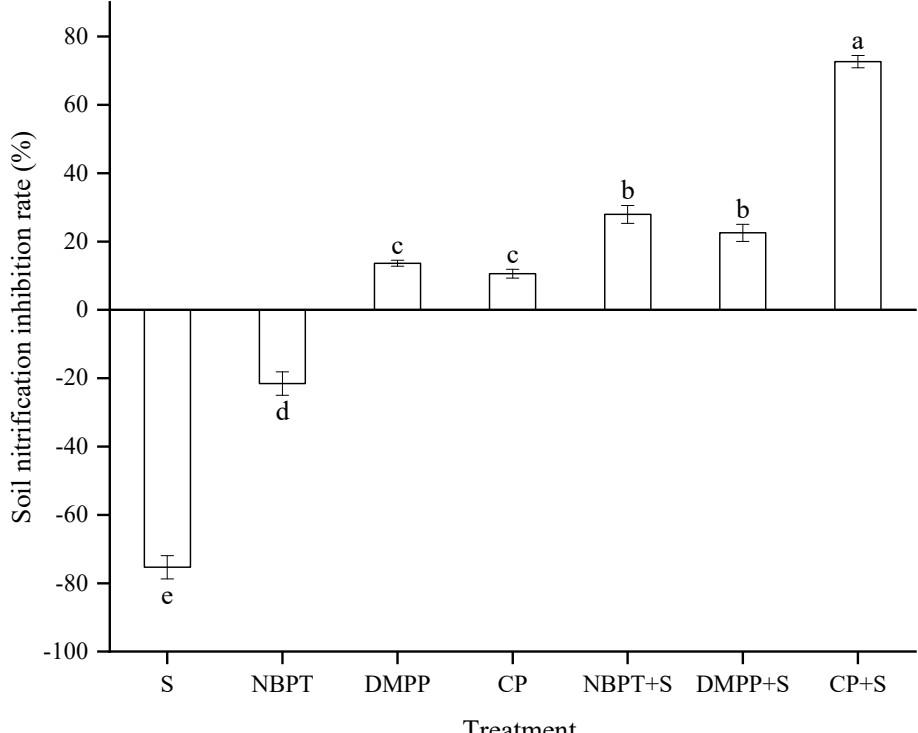

**Figure 1.** Soil nitrification inhibition rate of different treatments at the seedling stage. Error bars represented standard deviations ($n = 3$). Treatments: CK: no N fertilizer; U: urea; S: urea + seaweed extracts; NBPT: urea + N-butyl phos-phorothioate triamine; DMPP: urea + 3,4-dimethylpyrazole phosphate; CP: urea + 2-chloro-6-trimethylpyridine; NBPT + S: urea + N-butyl phosphorothioate triamine+ seaweed extracts; DMPP + S: urea + 3,4-dimethylpyrazole phosphate + seaweed extracts; CP + S: urea + 2-chloro-6-trimethylpyridine + seaweed extracts. Different letters indicate significant differences between different treatments at $p < 0.05$ by Tukey test.

### 3.3. The Changes in Enzyme Activity

#### 3.3.1. Soil Urease Activity of Different Treatments

The urease activity was low with each treatment at the seedling stage (Figure 2), and the combination of inhibitors and SE decreased the soil urease activities compared to applied inhibitor alone, among which CP + S significantly reduced the urease activity compared to CP ($p < 0.05$). At the elongation stage, the urease activities of the applied SE and inhibitors treatments were significantly higher than CK (Figure 2, $p < 0.05$). The soil urease activity showed an increasing trend with the growth of maize until the urease activity of each treatment was stabilized at the filling stage (Figure 2).

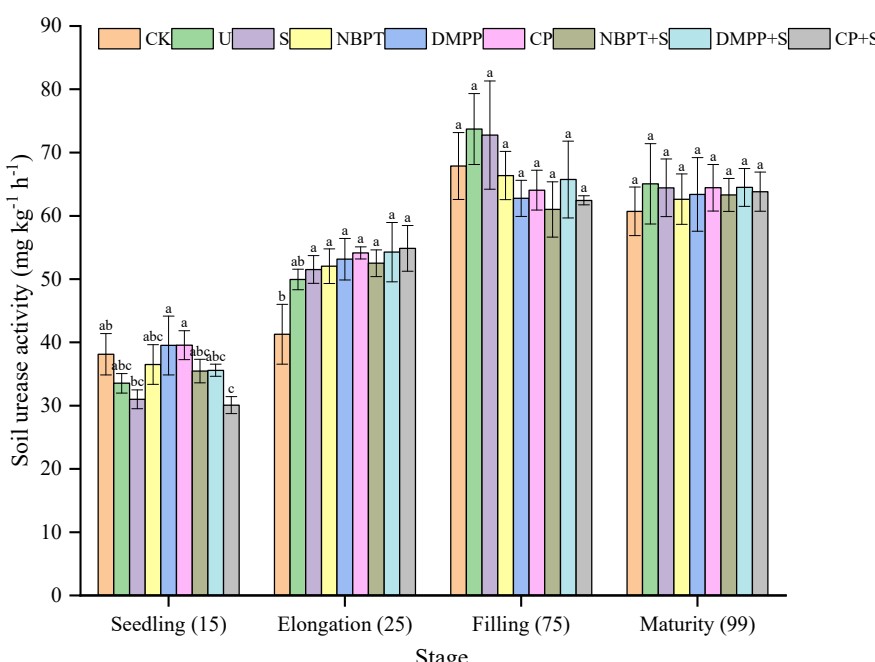

**Figure 2.** Soil urease activity of different treatments. Different letters indicate significant differences between different treatments at *p* < 0.05 by Tukey test.

### 3.3.2. Soil Potential Nitrification Rate (PNR) of Different Treatments

Soil PNR can indicate the rate of soil nitrification. At the seedling stage, the PNR of CP was significantly lower than U and DMPP, and CP was more effective than DMPP (Figure 3); the PNR of DMPP + S was significantly higher than DMPP; the PNR of CP + S was significantly higher than CP (*p* < 0.05); DMPP + S and CP + S were not beneficial to the effect of the 2 NIs. At the elongation stage, CP had the lowest PNR among the applied N treatments, indicating that CP had a longer-lasting effect, which was also in line with the change of $NH_4^+$-N contents (Table 2). From the seedling stage to the filling stage, the PNR of NBPT was maintained above 5.0 mg kg$^{-1}$ 5 h$^{-1}$, which was in line with the change of $NO_3^-$-N contents (Table 3). From the filling stage to maturity stage, the PNR of each treatment tended to be the same (Figure 3).

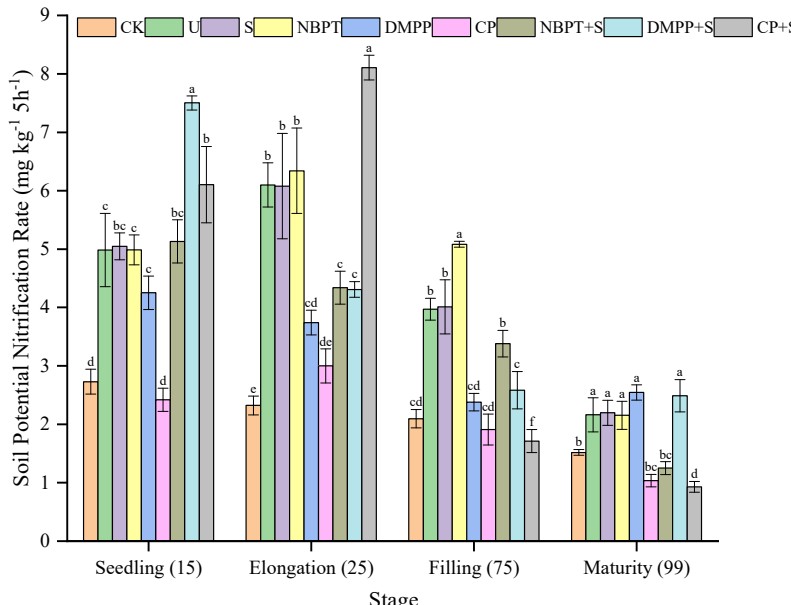

**Figure 3.** Soil Potential Nitrification Rate (PNR) of different treatments. Different letters indicate significant differences between different treatments at *p* < 0.05 by Tukey test.

### 3.3.3. Soil Nitrate Reductase (NR) Activity of Different Treatments

Maize is a rainfed plant and the soil is under aerobic conditions most of the time, thus the NR activity of each treatment is low overall. At the seedling stage, the NR activities of both S and NBPT were significantly higher than U ($p < 0.05$), and the application of both promoted soil denitrification (Table 3). The NR activity of CP was significantly higher than CP + S (Figure 4), which was in line with the change of $NO_3^-$-N contents (Table 3). At the elongation stage, the NR activity of each treatment was reduced; the NR activities of applied CP treatments were significantly higher than the other treatments. (Figure 4, $p < 0.05$). After the filling stage, the NR activity of all treatments tended to increase, and the NR activities of applied N treatments were higher than CK; at the maturity stage, the NR activities of the combination of inhibitors and SE were all significantly higher than those of applied inhibitor alone ($p < 0.05$), and the NR activities of the other treatments trended to be the same (Figure 4).

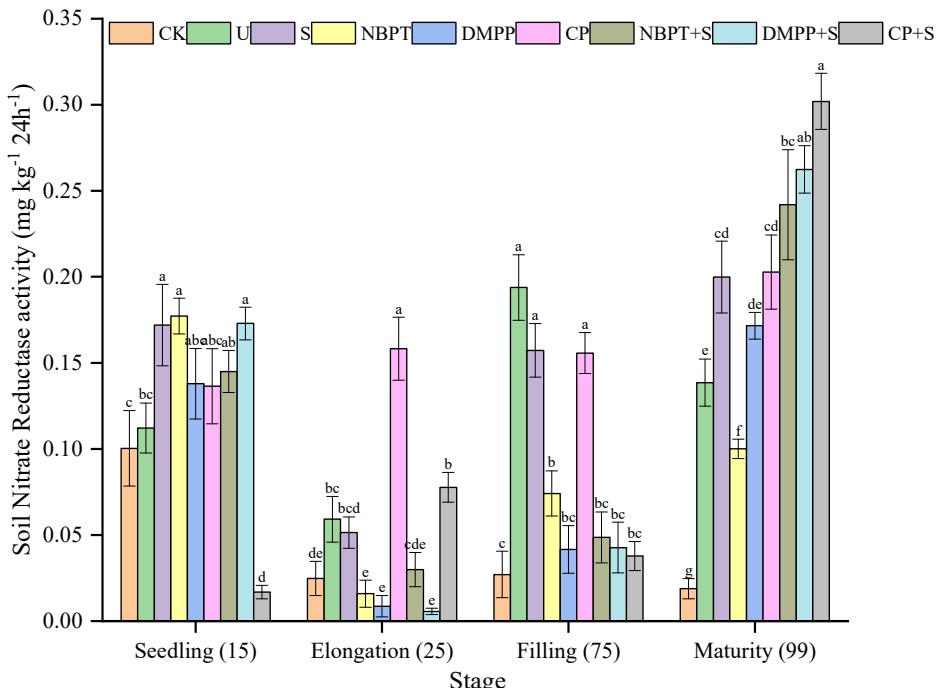

**Figure 4.** Soil Nitrate Reductase (NR) activity of different treatments. Different letters indicate significant differences between different treatments at $p < 0.05$ by Tukey test.

### 3.4. Soil Microbial Biomass Carbon (MBC) Contents of Different Treatments

Carbon (C) is an important component of microorganisms; thus, MBC contents can represent the microbiomass to some extent. From the seedling to elongation stage, the MBC contents of applied N treatments were all higher than CK (Figure 5). At the seedling stage, compared to the U, the applied of NBPT, DMPP, CP significantly reduced the MBC contents by 25.8%, 38.8%, and 40.2%, respectively ($p < 0.05$). At the elongation stage, the MBC contents of each treatment had increased, with the highest MBC contents with 216.2 mg kg$^{-1}$ of NBPT. At the filling stage, the MBC contents of applied N treatments were all lower than CK, which was in line with the change of $NH_4^+$-N contents (Table 2). At the maturity stage, the MBC contents of the combination of SE and inhibitors treatments were higher than U ($p < 0.05$), and the MBC contents of the applied SE and inhibitors treatments tended to be the same (Figure 5).

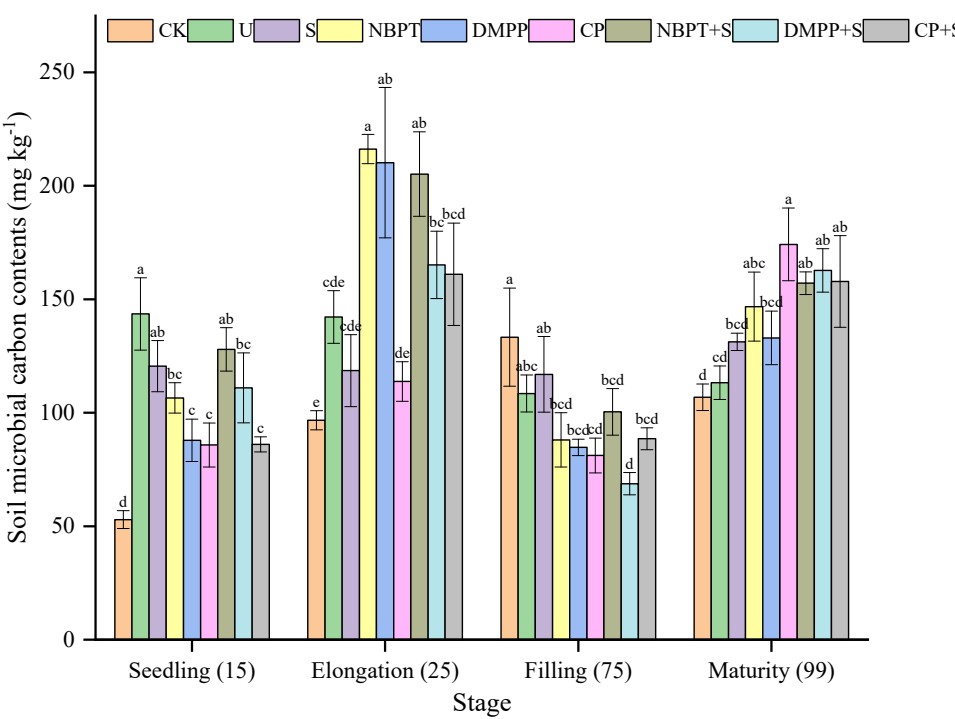

**Figure 5.** Soil microbial carbon (MBC) contents of different treatments. Different letters indicate significant differences between different treatments at $p < 0.05$ by Tukey test.

3.4.1. Maize Plant Physiological and Biological Indicators of Different Treatments

The N fertilizer application was positive for maize photosynthesis, chlorophyll content and leaf area of applied N treatments were significantly higher than CK (Table 4, $p < 0.05$). Compared to U, the treatments with the addition of inhibitors and SE increased the chlorophyll content and leaf area, among which CP had the best effect and all physiological and biological indexes of maize plants were high. There is no significant difference in chlorophyll content, leaf area, plant height and stalk thickness indexes between NBPT and NBPT + S; compared to DMPP, DMPP + S significantly increased the chlorophyll content by 29.1%; compared to CP, CP + S significantly reduced leaf area by 13.4% (Table 4, $p < 0.05$).

**Table 4.** Maize plant physiological at the silking stage (growth stage 67 in BBCH-scale) and biological indicators at maturity stage (growth stage 99 in BBCH-scale) of different treatments.

| Treatment | Chlorophyll | Leaf Area (cm$^2$) | Hight (cm) | Stalk Thickness (mm) |
|---|---|---|---|---|
| CK | 7.0 ± 0.3 d | 133.2 ± 2.6 d | 243.3 ± 13.5 b | 17.3 ± 1.6 b |
| U | 27.2 ± 3.7 c | 507.4 ± 24.9 c | 260.3 ± 3.8 ab | 22.0 ± 0.4 a |
| S | 44.7 ± 2.4 ab | 577.0 ± 17.5 bc | 263.3 ± 2.3 ab | 23.1 ± 1.7 a |
| NBPT | 50.6 ± 5.7 ab | 605.0 ± 30.8 ab | 259.0 ± 6.1 ab | 23.4 ± 2.0 a |
| DMPP | 40.6 ± 3.0 b | 638.8 ± 48.1 ab | 263.3 ± 8.5 ab | 23.7 ± 2.9 a |
| CP | 47.9 ± 6.2 ab | 684.6 ± 16.0 a | 270.0 ± 7.0 a | 25.4 ± 1.4 a |
| NBPT + S | 52.8 ± 2.4 a | 598.7 ± 40.6 b | 271.0 ± 3.6 a | 22.0 ± 1.3 ab |
| DMPP + S | 52.4 ± 2.5 a | 624.4 ± 19.6 ab | 275.7 ± 4.0 a | 21.4 ± 1.0 ab |
| CP + S | 53.6 ± 3.2 a | 592.8 ± 23.1 b | 257.0 ± 8.2 ab | 22.4 ± 1.3 a |

Different lowercase letters in the column mean a significant difference between treatments (Tukey test, $p < 0.05$). Treatments: CK: no N fertilizer; U: urea; S: urea + seaweed extracts; NBPT: urea + N-butyl phos-phorothioate triamine; DMPP: urea + 3,4-dimethylpyrazole phosphate; CP: urea + 2-chloro-6-trimethylpyridine; NBPT + S: urea + N-butyl phosphorothioate triamine + seaweed extracts; DMPP + S: urea + 3,4-dimethylpyrazole phosphate + seaweed extracts; CP + S: urea + 2-chloro-6-trimethylpyridine + seaweed extracts.

### 3.4.2. Maize Yield and Nitrogen Use Efficiency (NUE) of Different Treatments

Compared to U, the treatment with the addition of inhibitors and SE significantly increased total biomass, grain yield, plant N uptake and NUE (Table 5, $p < 0.05$). DMPP + S had positive application effect with the highest total biomass, plant N uptake and NUE of 399.5 g, 4.0 g and 89.1%, respectively. CP had the best application effect among the applied inhibitor alone treatments, with higher total biomass, grain yield, shoot/root ratio, plant N uptake and NUE than NBPT and DMPP, but root biomass was lower than NBPT and DMPP, indicating that CP application was not beneficial to the elongation of maize roots. Compared to NBPT, NBPT + S reduced the plant N uptake and NUE by 11.2% and 11.9%, respectively; compared to DMPP, DMPP + S significantly increased total biomass, plant N uptake and NUE by 17.7%, 28.5% and 30.6%, respectively; compared to CP, CP + S promoted the elongation of maize roots to some extent, and increased root biomass by 36.4% ($p < 0.05$).

**Table 5.** Maize yield, shoot/root ratio and nitrogen harvest index.

| Treatment | Total Biomass (g/Pot) | Grain Yield (g/Pot) | Root Biomass (g/Pot) | Shoot/ Root Ratio | Plant N Uptake (g/Pot) | NUE (%) |
|---|---|---|---|---|---|---|
| CK | 20.3 ± 2.6 d | 9.4 ± 1.0 c | 2.0 ± 0.3 c | 9.3 ± 2.5 d | 0.2 ± 0.0 e | - |
| U | 174.7 ± 7.8 c | 81.5 ± 3.2 b | 11.7 ± 1.4 b | 14.1 ± 2.3 cd | 2.2 ± 0.1 d | 47.0 ± 3.4 d |
| S | 365.7 ± 16.2 ab | 197.8 ± 10.6 a | 12.4 ± 2.3 b | 29.1 ± 5.7 ab | 3.5 ± 0.2 abc | 77.3 ± 4.1 abc |
| NBPT | 376.3 ± 22.9 ab | 176.7 ± 11.7 a | 17.2 ± 3.0 ab | 21.1 ± 2.3 abc | 3.8 ± 0.2 ab | 84.8 ± 4.1 ab |
| DMPP | 339.4 ± 7.8 b | 172.4 ± 9.1 a | 19.4 ± 2.7 a | 16.7 ± 2.0 cd | 3.1 ± 0.2 c | 68.2 ± 4.1 c |
| CP | 398.3 ± 22.9 ab | 200.0 ± 10.8 a | 12.4 ± 1.7 b | 31.7 ± 6.6 a | 3.6 ± 0.1 abc | 80.5 ± 1.9 abc |
| NBPT + S | 349.0 ± 21.5 ab | 172.1 ± 17.1 a | 15.2 ± 2.4 ab | 22.3 ± 3.1 abc | 3.4 ± 0.3 bc | 74.7 ± 7.0 bc |
| DMPP + S | 399.5 ± 35.7 a | 199.2 ± 23.1 a | 19.0 ± 3.0 a | 20.5 ± 4.6 bc | 4.0 ± 0.1 a | 89.1 ± 2.7 a |
| CP + S | 376.5 ± 25.5 ab | 186.8 ± 11.0 a | 16.9 ± 1.9 ab | 21.4 ± 2.0 abc | 3.6 ± 0.3 abc | 80.0 ± 7.9 abc |

Different lowercase letters in the column mean a significant difference between treatments (Tukey test, $p < 0.05$). Treatments: CK: no N fertilizer; U: urea; S: urea + seaweed extracts; NBPT: urea + N-butyl phos-phorothioate triamine; DMPP: urea + 3,4-dimethylpyrazole phosphate; CP: urea + 2-chloro-6-trimethylpyridine; NBPT + S: urea + N-butyl phosphorothioate triamine + seaweed extracts; DMPP + S: urea + 3,4-dimethylpyrazole phosphate + seaweed extracts; CP + S: urea + 2-chloro-6-trimethylpyridine + seaweed extracts.

## 4. Discussion

### 4.1. Effects of SE and Inhibitors on Soil Biochemical Parameters

In this study, the effect and time of inhibition from CP were better than DMPP in loess, which was according with Chen [45] in brown soil. According to Sanz-Cobena, the changes of enzyme activity depend largely on the substrate concentration [46], while the Urea-N was not founded in soil at the seedling stage, indicating that urea is fully hydrolyzed at this time, thus there is no significant difference in the urease activity between DMPP and CP (Figure 2, $p < 0.05$). There is no significant difference in $NO_3{}^-$-N contents between DMPP and CP (Table 3), and the reduction of nitrate to nitrite is catalyzed by NR, and its activity depends largely on the $NO_3{}^-$-N contents [47,48], thus there was no significant difference in NR activity between two treatments (Figure 4, $p < 0.05$). In general, microorganisms use $NH_4{}^+$-N as a more energy-efficient N assimilation pathway and strategy [49], however, the higher $NH_4{}^+$-N contents of CP compared to DMPP did not result in significant increase in MBC contents of CP to DMPP (Table 2, Figure 5, $p < 0.05$). The amount and form of nutrients in soil and the metabolites produced by the vital activities of roots could affect the growth of soil microorganisms [50]. In this study, DMPP has a better effect than CP in promoting the development of maize roots (Table 5), and the growth of maize roots was mainly concentrated in the seedling to elongation stage of maize [51], representing that DMPP had a more intensive root life activity at the seedling stage, that means more organic C would be input to soil through the root system [52], which provides a C source for the growth of microorganisms, and the metabolites produced by root also benefits the growth of microorganisms [53]. Therefore, despite the lower $NH_4{}^+$-N contents in soil of DMPP

(Table 2), there is no significant difference in soil MBC contents between DMPP and CP (Figure 5, $p < 0.05$).

SE significantly increased $NO_3^--N$ contents at the seedling stage compared to applied normal urea ($p < 0.05$). SE, as a complex substance with a high concentration of alginate, polysaccharides laminaran, and growth hormones, have all been shown to have positive bio-stimulating effects and increase the activity of soil microorganisms [26], which is beneficial for the acceleration of nutrient transformation in the soil. Compared to applied normal urea, the combination of SE and urea accelerated the transformation rate of Urea-N to $NH_4^+-N$ and then to $NO_3^--N$, increasing the concentration of $NO_3^--N$ in the soil in a shorter period of time (Table 3). While the alkaline loess in our experiment (Table 1) had strong loss of ammonia volatilization, SE shortened the retention time of $NH_4^+-N$ in the soil, thus reducing the loss of $NH_4^+-N$. Qiao et al. [54] showed that the addition of NIs increased ammonia volatilization by about 20%, and combined with the present experimental conditions, the intense ammonia volatilization loss in alkaline soils and the weaker denitrification under rainfed conditions both resulted in accelerated N loss when NIs were added, which is in line with the results of this experiment. Compared with DMPP and CP, S had higher soil inorganic N contents (Tables 2 and 3), meanwhile the experiment was set up for manual watering and the soil environment in the pots was relatively confined, which would reduce some of the $NO_3^--N$ leaching compared with field conditions, and the above factors resulted in significantly higher $NO_3^--N$ of S than the other treatments at the seedling stage, and the NR activity was significantly increased compared to U (Figure 4) due to the high concentration of $NO_3^--N$ (Table 3, $p < 0.05$).

Compared to applied inhibitor alone, the combination of SE and inhibitors decreased soil $NO_3^--N$ contents (Table 3), and the analysis of the results showed that the combination of inhibitors and SE could promote the uptake of $NO_3^--N$ by maize plants. The peak uptake capacity of $NO_3^--N$ was peaked between 40 and 65 days after maize planting [55], and previous results have shown that SE could accelerate the growth process of plants [56–58], and the results of this study showed that the combination of SE and inhibitors could accelerate the period of maximum nutrient uptake by maize, promoting the uptake of $NO_3^--N$ by maize at the seedling stage.

In terms of inhibitors effect analysis, NBPT + S significantly decreased $NO_3^--N$ contents compared to NBPT (Table 3), and the decrease in substrate concentration also caused a reduction in NR activity (Figure 4), and the various active substances and soluble C contained in SE also provided nutrients to soil microorganisms [53], resulting in an increase in the number of microorganisms at the seedling stage (Figure 5). Compared to DMPP, DMPP + S significantly increased PNR (Figures 3–5, $p < 0.05$), so the combination of DMPP with SE was not conducive to the nitrification inhibition effect of DMPP. DMPP is the dimethylpyrazol(DMP)-based NIs [59], soil microorganisms could break the phosphate group of DMPP, then DMP is released into the soil, which is the key substance to inhibit soil nitrification [60]. In this study, DMPP + S increased the MBC contents compared to DMPP (Figure 5), a result that leads to more intense microbial degradation of DMPP in the soil and shortens its effective action time. While the increase in the number of microorganisms to some extent marks the enhancement of soil enzyme activity [61], resulting in an increase of the soil nitrification intensity, which also reduces the nitrification inhibition effect of DMPP, and under the effect of both factors, the combination of DMPP with SE is not conducive to its action effect. Compared to CP, CP + S significantly reduced $NH_4^+-N$, $NO_3^--N$ contents, NR activity (Tables 2 and 3, Figure 4), but significantly increased PNR (Figure 3), while it did not cause a significant effect on MBC contents (Figure 5, $p < 0.05$). The decrease in soil inorganic N contents of CP + S was caused by the increase of root biomass of maize plants, and the decrease in $NO_3^--N$ contents also caused a significant decrease of NR activity (Figure 4, $p < 0.05$). Meanwhile, analyzing with all indicators, the combination of CP and SE reduced the effect of CP. Due to the presence of an unshared electron pair on the N atoms in chloropyridine, which is the hydrolysis product of CP, the carboxyl groups of soil humic materials could bind to the unshared electron pair of chloropyridine through electrostatic

bonding, and thus reducing the effect of CP, while carboxyl groups are also present in SE [62], resulting in a reduction in the effect of CP when CP are combined with SE.

*4.2. Effects of Inhibitors and SE on Physiological and Biological Indexes of Maize Plants and NUE*

In this study, compared to the normal urea, adding SE, inhibitors and their combinations all increased plant N uptake, then increased chlorophyll content, leaf area, yield, and NUE (Tables 4 and 5), which is in accordance with the findings of the previous studies [63–66]. Engel et al. [67] showed that the half-life of NBPT in alkaline soil was only 3.43 days. While Soares et al. [68] showed that in brown soils, NBPT was effective in inhibiting soil urease activity only one week after application, meanwhile the combination of NBPT with SE increased the number of soil microorganisms at the seedling stage, indicating that the addition of SE further shortened the effective time of NBPT, which is not conducive to the effect of NBPT, while SE could promote plant and soil microorganism growth, which has a positive effect on plant yield formation [24], thus there was no significant difference in maize physiological and biological indexes between NBPT and NBPT + S ($p < 0.05$). Compared to DMPP, DMPP + S significantly increased the plant N uptake, then increased chlorophyll content, total biomass and NUE (Tables 4 and 5, $p < 0.05$), and the combination of DMPP and SE had a positive effect, the combination of DMPP with SE was not conducive to the effect of DMPP, but the combination of both resulted in a positive yield increase. Sidhu et al. [69] showed that DMP, as the degradation product of DMPP, is degraded in soil mainly by soil chemical processes and not by direct microbiological activity, thus the addition of SE could promote the decomposition of DMPP in soil, but it did not have a large effect on the degradation of DMP, therefore DMP could still maintain the inhibition effect of soil nitrification for a longer period of time, meanwhile, the loess used for this experiment had a looser soil texture and thus DMPP effect was better [70]. The negative effect of the combination of DMPP with SE in loess was much less than the positive effect of the addition of SE, and the combination of DMPP with SE was beneficial to yield improvement. Compared to CP, CP + S promotes the development of the maize root system and had no significant effect on plant N uptake, yield, and NUE (Table 5, $p < 0.05$), and in the above discussion, CP combined with SE could reduce CP effectiveness and greatly promote maize root development (Table 5), which was also beneficial to the uptake of soil nutrients by maize plants. Although CP combined with SE was not conducive to CP effect, it did not cause significant effects on yield and NUE.

Through analyzing the factors above, we concluded that the combination of inhibitors and SE will reduce the effect of inhibitors application, however, maize yield and NUE are the most important standards for judging fertilizer formulation, so we consider that the combination of DMPP and SE still has good application effect.

## 5. Conclusions

Compared with normal urea, the addition of SE, inhibitors and their combinations had different effects on the transformation characteristics of Urea-N in loess, which could effectively promote maize growth and significantly increase yield and NUE. In loess, compared to the application of NBPT, DMPP and CP alone, the combination inhibitors with SE were all not conducive to their effectiveness, among which NBPT + S decrease the maize yield and NUE to some extent; DMPP + S can significantly increase the total biomass, plant N uptake and NUE; CP + S could promote the growth and development of the maize root system ($p < 0.05$). The combination of inhibitors and SE is beneficial to the uptake of soil N by plants during the early stages of maize growth. The effect of SE on promoting the maize growth and enhancing the uptake of soil $NO_3^-$-N by maize can be an effective strategy to reduce the loss of soil $NO_3^-$-N. Among all fertilizer formulations, the combination of DMPP with SE has the best application effect and the highest maize yield and NUE.

**Author Contributions:** Conceptualization, F.X. and D.L.; Data curation, F.X., P.G., Y.S.; Formal analysis, F.X.; Funding acquisition, D.L. and L.Z.; investigation, F.X., D.L., Y.D., Y.X., K.Z., Y.Z., Y.L., J.Z. and Y.C.; methodology, F.X.; Resources, L.Z., Y.D., Y.X., L.C., P.G., Y.S., K.Z., Y.Z., Y.L., J.Z. and

Y.C.; Supervision, D.L.; Writing—original draft, F.X.; Writing—review and editing, F.X. and D.L. All authors have read and agreed to the published version of the manuscript.

**Funding:** This work was funded by the Strategic Priority Research Program of the Chinese Academy of Sciences (XDA28090200), the National Key Research and Development Program Project of China (2017YFD0200707), the National Scientific Foundation Project of China (31971531).

**Institutional Review Board Statement:** Not applicable.

**Informed Consent Statement:** Not applicable.

**Data Availability Statement:** All relevant data is contained within the article.

**Acknowledgments:** The National Field Research Station of Shenyang Agroecosystems, Chinese Academy of Sciences, for providing the experimental field.

**Conflicts of Interest:** The authors declare no conflict of interest.

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
