# Peer review of "Effect of Urease Inhibitors and Nitrification Inhibitors Combined with Seaweed Extracts on Urea Nitrogen Regulation and Application"

_agronomy, doi:10.3390/agronomy12102504_

Round 1

Reviewer 1 Report

Manuscript ID: agronomy – 1948477

A.      Main comments

1.      The abstract needs to be corrected.

A properly constructed abstract contains: key concept(s), basic aspects of materials, results and conclusions that summarize not only the current state of knowledge, but also address new solutions. The length of the abstract is of 200 words.

2.      Keywords cannot coincide with title of the article.

3.      The introduction contains a number of side elements. However, important elements concerning the operation of N inhititors have been omitted.

4.      The objectives of the work require re-editing so that the reader gets the essence of the scientific problem. 5.      What was the basis for the nitrogen dose being lower than the P and K dose? Such a small dose could have a significant impact on the obtained test results. In this type of experiment there should be at least two doses of nitrogen !!! 6.      The obtained results for each of the tested characteristics should be compared with the combination with nitrogen (only urea), and not with the absolute control. 7.      All Figures should be supplemented with the development phase expressed in the BBCH scale. 8.      Figures in tables contain too many characters, which reduces their legibility. 9.      The readability of the tables is very poor. 10.  Each table must describe the fertilizer combinations. 11.  The discussion should be reworked. In this type of work, the discussion should begin with the end result, i.e. grain yield. It is the end result that determines the value of the tested fertilizer prototypes. Traits with little efefct on yeild cannot be discussed widely. Detailed comments are included directly into the text.

Reviewer 2 Report

Dear authors,

Thank you for the opportunity to review this Manuscript (Effect of urease inhibitors and nitrification inhibitors combined with seaweed extracts on urea nitrogen regulation and application). The study has great results and demonstrates that the addition of SE was beneficial to N uptake by plants at the early growth stage of maize, but reduced the inhibitors effect, where the DMPP+SE, CP+SE significantly increased the potential nitrification rate. Concluding that the addition of DMPP 38 with SE to urea significantly improved yield when planting maize in loess areas. There is some aspect that should be reviewed by authors, but the Manuscript is well-written.

INTRODUCTION

Lines 58-73: authors could explore the process of inhibitors in soil

Explain the difference between the inhibitors, where each one act in N cycle

Line 68, Explain the difference between the inhibitors in the soil texture

The goals are no clear. Please, check it.

MATERIAL AND METHODS

Explain, what is “Surface loess samples” and how/where it was collected

Give more details about the pots/vases (size, for example)

Add the weight of soil used

Explain YMJ-B and CCM-200. What is?

RESULTS AND DISCUSSION

Correlation between NH4+ and NO3- could demonstrate by the authors.

Is there information of soil pH? If yes, the data could be added.

The high Soil nitrification inhibition rate of CP+S should be explained.

The urease activity in seedling also was caused by lower plant N adsorption?

The discussion and conclusion are very good.

Round 2

Reviewer 1 Report

 A.      Main comments

1.      The abstract needs to be corrected.

A properly constructed abstract contains: key concept(s), basic aspects of materials, results and conclusions that summarize not only the current state of knowledge, but also address new solutions. The length of the abstract is of 200 words.

2.      The hypothesis cannot refer to soil type. Loess soils are found in many regions of the world: Russia, Ukraine, Germany, USA, Canada, for example.

3.      What was the basis for the nitrogen dose being lower than the P and K dose? Such a small dose could have a significant impact on the obtained test results. In this type of experiment there should be at least two doses of nitrogen !!! 4.      The obtained results for each of the tested characteristics should be compared with the combination with nitrogen (only urea), and not with the absolute control. 5.      The discussion should be reworked. In this type of work, the discussion should begin with the end result, i.e. grain yield. It is the end result that determines the value of the tested fertilizer prototypes. Traits with little effect on yield cannot be discussed widely. The reader is looking forward to an argument that confirms the advantage of DMPP + S over other experimental variants.

6.      The conclusions are inconsistent with the presented research results.

  Detailed comments are included directly into a manuscript.
